# Reflex Locomotion Therapy for Balance, Gait, and Fatigue Rehabilitation in Subjects with Multiple Sclerosis

**DOI:** 10.3390/jcm11030567

**Published:** 2022-01-23

**Authors:** María Carratalá-Tejada, Alicia Cuesta-Gómez, Rosa Ortiz-Gutiérrez, Francisco Molina-Rueda, Laura Luna-Oliva, Juan Carlos Miangolarra-Page

**Affiliations:** 1Motion Analysis, Ergonomics, Biomechanics and Motor Control Laboratory (LAMBECOM), Department of Physical Therapy, Occupational Therapy, Rehabilitation and Physical Medicine, Faculty of Health Sciences, Rey Juan Carlos University, 28922 Madrid, Spain; maria.carratala@urjc.es (M.C.-T.); alicia.cuesta@urjc.es (A.C.-G.); laura.luna@urjc.es (L.L.-O.); juan.miangolarra@urjc.es (J.C.M.-P.); 2Department of Physical Therapy and Nursing, Universidad Internacional Nebrija, 28015 Madrid, Spain; rortizg@nebrija.es; 3Physical Medicine and Rehabilitation Service, University Hospital of Fuenlabrada, 28942 Madrid, Spain

**Keywords:** gait analysis, kinematics, kinetics, multiple sclerosis, Vojta therapy, reflex locomotion therapy

## Abstract

This study evaluates the effects of a rehabilitation program based on reflex locomotion therapy (RLT) on balance, gait, and fatigue in patients with multiple sclerosis (MS). Twenty-three patients diagnosed with MS participated in this study. Reversal design was carried out. The assessment tools included the Berg Balance Scale (BBS), the Performance Oriented Mobility Assessment (POMA), the Fatigue Severity Scale (FSS) and the instrumental analysis of the gait recorded by Vicon Motion System^®^. We analyzed spatio-temporal parameters and kinematic variables of the hip, knee, and ankle joints. Additionally, the Client Satisfaction Questionnaire (CSQ-8) was administrated. We did find a significant improvement in balance and gait tools after the RLT period. Regarding instrumental analysis, the statistical analysis of spatio-temporal parameters showed a significant improvement in stride length, double support, and velocity after the RLT period. Concerning kinematic parameters, the analysis showed improvements in hip and knee range of motion (ROM) after RLT period. RLT could improve gait and balance in patients with MS. The patients reported a high level of satisfaction with the therapy received.

## 1. Introduction

Multiple sclerosis (MS) is a progressive disease with a fluctuating and unpredictable course. The disease is characterized by the accumulative effect of multiple injuries at numerous levels. These variables depend on each patient and their stage of the disease. The disabling nature of MS leads to a decline in quality of life, with considerable repercussions affecting the social environment [1].

MS can cause sensory (40%), pyramidal (40%), cerebellar (25%), and visual symptoms (20%) [2]. Visible symptoms include the use of assistive devices, problems with balance, and speech difficulties, while invisible symptoms include fatigue, pain, depression, and anxiety [3]. The level of disability is scored using the Expanded Disability Status Scale (EDSS), a useful scale for measuring neurological disability (pyramidal, cerebellar, brainstem, sensory, bowel, bladder, visual, and cerebral functions) [4]. Among the variability of signs and symptoms of MS, the motor disorders disturbing balance, postural control (PC) and gait are among the most common. Thus, it is considered that a certain level of recovery regarding these disorders is associated with an overall improvement in the functional autonomy and quality of life [5].

Pharmacological treatments are not always effective, therefore, several authors [6,7] highlight the importance of rehabilitation therapy, and specifically physical therapy, as a main treatment tool. According to Lord et al. [8], a short period of ambulatory physical therapy (four to seven weeks) can significantly influence the physical capacities of patients with MS.

Several physical rehabilitation methods are available for the treatment of balance and gait in subjects with multiple sclerosis such as task-oriented training, sensory-motor integration training, reflex locomotion therapy (RLT) or conventional dynamic strengthening exercises [9]. In this sense, RLT is a method of movement therapy that activates the pre-organized circuits of the central nervous system (CNS), triggering motor programs with locomotion components. Keeping a specific posture and stimulating specific anatomical points located in the musculature could stimulate the CNS through proprioceptive afferents. This information triggers the innate response of the locomotor coordination complexes that are used by human during walking [10]. Recently, Sanz-Esteban et al., (2021) [11] demonstrated that there were responses at the cortical level to a specific tactile input induced with RLT. The same authors reported that RLT activates innate muscle responses using surface electromyography [12]. As noted by Laufens et al. [13,14], RLT could be an effective therapy to improve the motor function in patients with MS. These authors showed significant improvements in gait speed and step length. Works that are more recent have shown relevant modifications in gait and balance in patients with MS using observational scales [15,16]. However, studies are necessary to evaluate the effects of RLT on balance and gait in people with MS using objective instruments such as three-dimensional movement analysis systems. Thus, the aim of the present work was to evaluate the effects of a treatment based on RLT to improve balance, gait, and fatigue in patients with MS. In addition, we evaluate the satisfaction of health services users.

## 2. Materials and Methods

### 2.1. Participants

We requested the voluntary participation of subjects with MS. The patients had to meet the following inclusion criteria: aged between 20 and 60 years; diagnosed with MS of over two years’ evolution based on the McDonald criteria [17]; receiving rehabilitation treatment for at least the previous two years; Expanded Disability Status Scale (EDSS) scores of between 4 and 6; absence of cognitive decline, with the ability to understand instructions, with a score of ≥20 on the Mini-Mental State Examination (MMSE). We excluded subjects who over the previous six months and during the research had suffered a worsening of symptoms, had required hospitalization, corticoid therapy, either intravenous or oral, botulinum toxin, or experienced any other situation that could potentially hamper their participation in the study.

All subjects included in the present study were informed of the objectives, protocol and risks of and voluntarily accepted to participate, providing their consent in writing. The protocol, the information provided to the patient and the informed consent were approved by the Research Ethics Committee of the Rey Juan Carlos University.

### 2.2. Sample Size Calculation

The effect size estimated for the main assessment measures established in the present work was 0.30. Considering a statistical power of 0.90; an alpha error of 0.05 and a correlation between the repeated measures of 0.5 (one group, four measurements); a minimum of 22 subjects were required for the present study, according to Software G*power (version 3.1.7).

### 2.3. Study Procedure

The experimental protocol was designed based on a reversal design (Single-subject research), also called the ABA, with three intervention periods: (A) conventional physiotherapy, (B) RLT and conventional therapy and (A) withdrawal of RLT. Each condition lasted for six weeks (Figure 1).

Regarding the intervention, during the period A, the patients received their conventional ambulatory physiotherapy treatment at their respective associations, twice weekly, based on active and passive movement therapy, musculotendinous stretching and balance and gait training.

During the period B, the patients began the RLT protocol. The recruited subjects were treated twice a week, on non-consecutive days, over a six-week period, receiving 12 sessions in total. The treatment sessions consisted of 40 min RLT plus 20 min conventional therapy.

The assessments and instrumental analysis were performed, on the same day, before and after each condition (four measurement times). After the established intervention period, the patients completed a satisfaction questionnaire regarding the intervention (CSQ-8).

The instrumental analysis of the gait was recorded by Vicon Motion System^®^ (Vicon Motion Systems, Oxford, UK) using 8 MX 13+ infrared capture cameras and three AMTI^®^ (Accent Micro Technologies Inc., Watertown, MA, USA) dynamometric force platforms (410 × 610 mm), located in the middle of an 11 m walking corridor.

For instrumental gait analysis, passive and reflective markers were placed in specific anatomical areas of the lower limbs (anterior superior iliac spine, posterior superior iliac spine, middle third of thigh, external femoral condyle, middle third of tibia, external malleolus, calcaneus, and head of second metatarsal), according to the biomechanical models of Davis et al. [18] and Kadaba et al. [19]. After instrumentation, the subjects were asked to walk at a self-selected comfortable gait speed by corridor lab, recording five repetitions per subject in each session.

All the measurements were performed at the Movement Analysis, Biomechanics, Ergonomics and Motor Control Laboratory (LAMBECOM) located at the Faculty of Health Sciences (Rey Juan Carlos University).

### 2.4. Intervention

A rehabilitation program was administered based on RLT. Reflex locomotion presents two patterns, known as reflex crawling and reflex rolling [20]. These patterns of reflex locomotion are innate and can be accessed from specific positions and by pressing upon specific points in certain directions. Four positions were performed on all patients (Figure 2), two of which were reflex crawling (original position of reflex crawling and first position), while two were reflex rolling (first phase and second phase of rolling), for as long as the patient could tolerate these; otherwise, small modifications were made, ensuring the comfort of the individual involved. The stimulation points activated in the position of reflex crawling were mainly the medial epicondyle of the facial arm, the external protuberance of the calcaneus of the nuchal leg, the anterior superior iliac spine (ASIS) of the facial hemipelvis and the femoral condyle medial to the nuchal side. During the reflex rolling, the main points stimulated were, in supine lying, the 6th-7th intercostal space and the ASIS of the hemipelvis on the facial side; and, in side lying, the internal border of the scapula, the ASIS of the facial hemipelvis, the facial gluteus medius, the femoral condyle external to the nuchal side, the internal femoral condyle of the facial side and the external protuberance of the calcaneus of the nuchal side.

The therapy was always administered by the same therapist who was a physiotherapist trained in RLT.

### 2.5. Outcome Measures

#### 2.5.1. Berg Balance Scale

The Berg Balance Scale (BBS) [21] was developed as a quantitative measure of the functional status of balance for the older person. The total score ranges between 0 (severely affected balance) to 56 (excellent balance). In this sense, several authors [22,23,24] have confirmed that this scale presents appropriate levels of validity and reliability for the assessment of balance in patients with MS.

#### 2.5.2. Performance Oriented Mobility Assessment

The Tinetti Balance and Gait test or the Performance Oriented Mobility Assessment (POMA) was developed to detect balance and mobility problems in older people [25]. This test is formed by two subscales, one for balance and another for gait. The sum of both equals a total score of 28. Different authors [26,27] suggest that the high precision of POMA makes it well suited for assessing balance disorders in patients with MS.

#### 2.5.3. Biomechanical Parameters

The spatio-temporal parameters analyzed in this study were stride length, single and double support, and the velocity. Additionally, we analyzed the motion of the hip, knee, and ankle in the sagittal plane. The following kinematic parameters were analyzed: hip range of motion, hip angle at initial contact, peak hip extension during stance period, peak hip flexion during swing period, knee range of motion, knee angle at initial contact, peak knee flexion during swing period, ankle range of motion, ankle angle during the initial contact and peak ankle plantarflexion at toe off.

#### 2.5.4. Assessment of Fatigue

The Fatigue Severity Scale (FSS) was described by Krupp et al. in 1989. This consists of nine items which are assessed by the patient with a score of between 0 and 7: the higher the score, the greater the fatigue [28].

#### 2.5.5. Assessment of Satisfaction

The Client Satisfaction Questionnaire (CSQ-8) is designed to evaluate the satisfaction of health services users [29]. The model is a self-administered post-treatment questionnaire, comprising eight items which, among other aspects, assesses the level of satisfaction regarding the care and quality of the service received and the level of fulfillment of the patient’s expectations regarding the treatment administered.

#### 2.5.6. Data Analysis

The Vicon^®^ Nexus software v1.8.5 was used to calculate outcome measures based on the biomechanical model of the Vicon^®^ Plug-in Gait. The output angles for all joints were calculated from the YXZ cardan angles derived by comparing the relative orientations of the two segments. The position of the hip segment was relative to the proximal segment, i.e., the hip to the pelvis. The course and direction of the segment axes are shown in the Vicon^®^ Plug-in Gait Product Guide-Foundation Notes Revision [30].

Five gait cycles were averaged for the data analysis. The foot contact events were defined automatically, using the “autocorrelation events” option of the Vicon^®^ Nexus software v1.8.5 according to the vertical force component measured by the force plates [31].

#### 2.5.7. Statistical Analysis

The SPSS (version 22.0) statistical package was used for analysis. The results were expressed as the mean, difference of means (DifM), standard deviation (SD) or 95% confidence interval (95% IC). The Kolmogorov-Smirnov test demonstrated that the values of the variables showed a normal distribution (*p* > 0.05). An ANOVA of repeated measures with Bonferroni adjustment was performed to compare the parameters between each of the measures. The statistical analysis was performed with a 95% confidence interval and the significant values were those with a *p* of <0.05. Effect size was calculated using the Cohen’s d.

## 3. Results

The sample comprised a total of 23 patients (EDSS between 4 and 6), of whom 27 were initially assessed. There were four dropouts: two patients get flare-ups, one patient became pregnant, and another was unable to attend treatment due to logistical problems. No deleterious effect was registered during the study. Table 1 presents the description of the sample, regarding the quantitative and qualitative variables.

The statistical analysis showed a significant improvement in balance scales after the RLT period (BBS: *p* < 0.001; CI95% 1.67–5.37; Cohen’s d = 0.54); POMA balance subsection: *p* < 0.001; CI95% 0.99–2.75; Cohen’s d = 0.77). Additionally, the POMA gait subsection also improved after the RLT period (*p* < 0.001; CI95% 0.81–1.98; Cohen’s d = 0.61). The FSS did not display a significant variation between the assessed times (Table 2 and Figure 3).

The statistical analysis of spatio-temporal parameters showed a significant improvement in stride length (*p* < 0.01; CI95% 0.05 to 0.18; Cohen’s d = 0.46), double support (*p* = 0.047; CI95% −0.12 to −0.0004; Cohen’s d = 0.25) and velocity (*p* = 0.001; CI95% 0.04 to 0.19; Cohen’s d = 0.43), after the RLT period (Table 3). Regarding kinematic parameters, the analysis showed a variation in hip and knee joints, but not in ankle joint (Table 4). At the hip joint, significant differences were observed in ROM (*p* = 0.006; CI95% 0.56 to 4.77; Cohen’s d = 0.35) between second and third measurement. At the knee joint, we observed significant changes after the RLT period (ROM: *p* = 0.045; CI95% −0.40 to 6.43; Cohen’s d = 0.22).

The analysis of the CSQ-8 showed, a high degree of satisfaction, in general, obtaining a mean of 30.65 points out of the maximum of 32. Of the eight items that this questionnaire considers, the maximum score was achieved for the whole sample in response to questions 1 (How do you evaluate the quality of the service you received?), 5 (Are you satisfied with the help you have received?) and 7 (In general, are you satisfied with the services you have received?). None of the participants showed any disagreement or dissatisfaction with any of the remaining questions posed.

## 4. Discussion

RLT has been applied in adults with neurological conditions during several decades. Belda et al. [32] support the application of RLT as one of the most used European physiotherapy techniques in neurological patients, and several authors [12,13,14,15] use RLT specifically in patients with MS. 

In this study, we found significant differences in balance using the BBS and the POMA after the RLT intervention. Cohen´s d informed about a medium effect size in the BBS and the POMA improvements. Regarding the assessment of the impact of RLT on fatigue, our results showed no significant differences between the assessment periods. The participants showed a high level of satisfaction with the physiotherapy treatment received.

Imbalance and falls are common in people with MS. In all, 50-80% have balance, and gait dysfunction and over 50% fall at least once each year [33]. In the present work, the differences obtained after the RLT intervention in balance (3.52 points using the BBS) are higher that the changes observed in recent studies that evaluated other interventions for balance rehabilitation (1.9–2.6 points using the BBS) [9]. Our results are congruent with Guner and Inanici [34], which showed adequate results in balance and spatio-temporal parameters, after applying a yoga program in patients with MS, using the BBS and a three-dimensional gait analysis system. In this sense, to achieve a better balance performance could improve the participation and the quality of life in MS patients. 

In our study, we also found significant improvements in the stride length and velocity after the RLT period. Several studies have showed a reduced speed of progression, shorter strides, and prolonged double support intervals in MS patients [35,36]. Preiningerova et al. [36], observed that speed decreases as the level of disability increases according to the EDSS. Therefore, to obtain an improvement in speed and stride length in people with MS is a relevant finding that could improve the functionality of these patients. In addition, we showed significant modifications in gait kinematics after the RLT period. Specifically, participants with MS improved hip and knee ROM by 3 degrees. This magnitude of change is above the two degrees of error that is considered acceptable for three-dimensional motion analysis systems [37]. However, these modifications showed a small effect size according to the Cohen’s d. Regarding knee joint, impaired knee range of motion could be one of the main kinematic features of MS gait [2]. The reduced knee motion during gait may be caused by paresis or increase in muscle tone (e.g., hamstrings and rectus femoris), or decreased push-of power in the ankle joint (e.g., gastrocnemius) [38].

The RLT has demonstrated positive results in studies that included adults. Pavlu et al. [39] demonstrated changes in respiratory frequency and trunk muscle activation after a RLT intervention. Recently, Ha and Sung [40] observed that the stimulation of the pectoral muscle point during reflex rolling in healthy subjects improve the activity of the transversus and the diaphragm muscles. Regarding patients with neurological disorders, Husárová [41] in patients with hemiparesis, reported significant improvements in gait, spasticity, and the articulation of the speech. Perales et al. [42] showed modifications in the gait pattern of stroke patients using the Wisconsin Gait Scale and the 10m walking test. Specifically, Laufens et al. [13,14,16] in patients with MS, described improvements in gait speed, step length, surface electromyography and EDSS score, after the application of RLT. Finally, some authors have demonstrated that RLT increase the automatic postural reactions and improve balance in patients with neurological conditions [15,43]. 

The present study shows some limitations that should be improved in future studies. The reduced sample size and the study design without control group make difficult the interpretation of the results. In this sense, we cannot clarify whether the results obtained are an accumulative effect of the training periods. Further works should consider other designs with a control group. We did not distinguish between the different forms of presentation and the motor clinical characteristics of the patients, including both patients with an ataxic, spastic, or mixed component. In addition, there is a risk for bias, as we were unable to cease the conventional physiotherapy treatment during the assessment periods, due to obvious ethical reasons. Therefore, we cannot ensure that the improvements obtained were solely due to RLT. Finally, the intervention was performed with subjects who had a score of between four and six on the EDSS and, therefore, our results cannot be extrapolated to other degrees of involvement.

## 5. Conclusions

Subjects with MS after a RLT intervention improved balance, walking spatio-temporal parameters (velocity, double support, and stride length) and walking kinematics (hip and knee range of motion). The participants have a high level of satisfaction with the intervention received according to the Client Satisfaction Questionnaire.

## Figures and Tables

**Figure 1 jcm-11-00567-f001:**
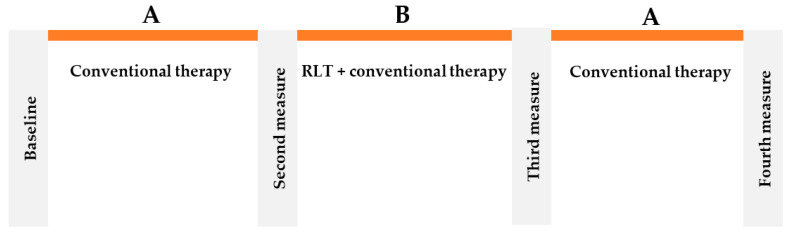
Experimental procedure.

**Figure 2 jcm-11-00567-f002:**
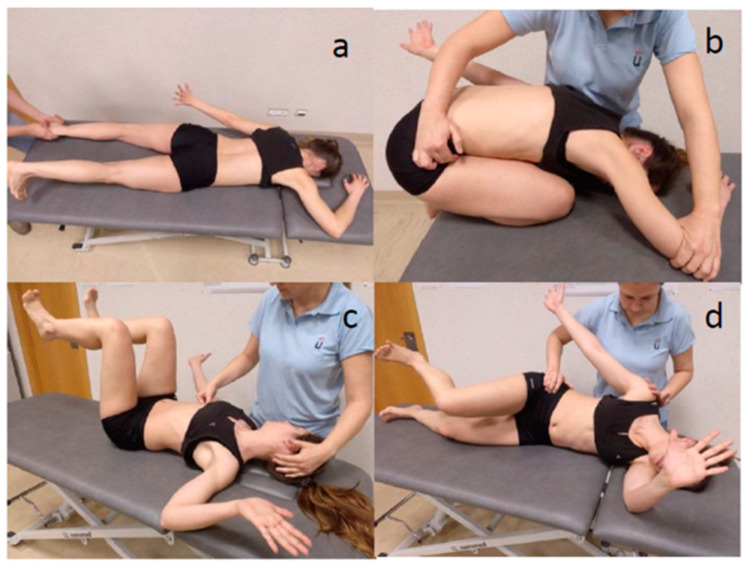
Application of reflex locomotion therapy (RLT). Reflex crawling: (**a**) original reflex crawling and (**b**) first position; reflex turning: (**c**) first and (**d**) second phase of turning.

**Figure 3 jcm-11-00567-f003:**
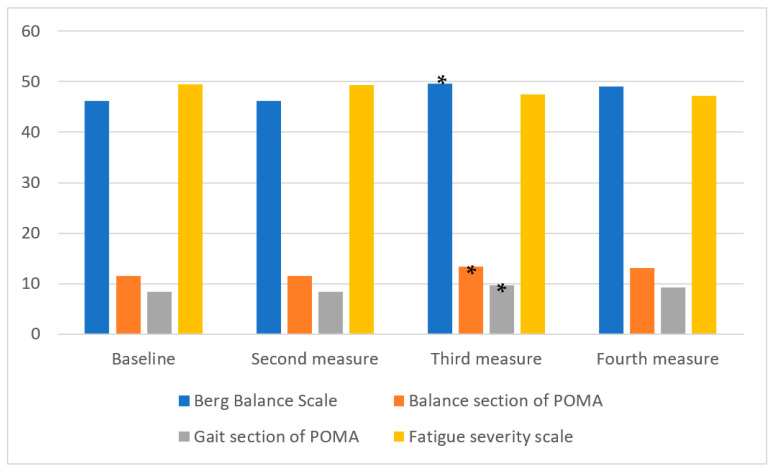
Change (mean) in balance, gait and fatigue parameters during the experimental protocol. * *p* < 0.05 using ANOVA of repeated measures between the third and second measurements.

**Table 1 jcm-11-00567-t001:** Characteristics of the sample.

Variable	Subjects		
Age (years) *	44.39 (7.37)		
Sex	♀ 73.91% (*n* = 17)		♂ 26.08% (*n* = 6)
Most affected side	Right 43.48% (*n* = 10)		Left 56.56% (*n* = 13)
Years since diagnosis *	14.39 (5.85)		
Type of MS	RRMS 56.52% (*n* = 13)	SPMS 21.73% (*n* = 5)	PPMS 21.73% (*n* = 5)

* Data expressed as mean (standard deviation). MS. Multiple Sclerosis. EMRR. Relapsing remitting multiple sclerosis. SPMS. Secondary progressive multiple sclerosis. PPMS. Primary progressive multiple sclerosis. ♀: woman. ♂: men.

**Table 2 jcm-11-00567-t002:** Clinical variables: tests for balance, gait and fatigue.

	Mean (SD)				Second vs. Baseline Measurement			Third vs. Second Measurement			Fourth vs. Third Measurement		
	Baseline	Second Measure	Third Measure	Fourth Measure	DM	*p*	CI 95%	DM	*p*	CI 95%	DM	*p*	CI 95%
Berg Balance Scale	46.21 ± 7.24	46.13 ± 7.17	49.65 ± 5.56	49.08 ± 6.26	−0.09	1.000	−1.94 to 1.76	3.52	<0.001 *	1.67 to 5.37	−0.57	1.000	−2.41 to 1.28
Balance section of POMA	11.56 ± 2.51	11.47 ± 2.71	13.34 ± 2.08	13.13 ± 1.79	−0.09	1.000	−0.97 to 0.79	1.87	<0.001 *	0.99 to 2.75	−0.22	1.000	−1.10 to 0.66
Gait section of POMA	8.43 ± 2.33	8.39 ± 2.38	9.73 ± 1.98	9.30 ± 2.22	−0.04	1.000	−0.58 to 0.50	1.35	<0.001 *	0.81 to 1.89	−0.43	0.190	−0.97 to 0.10
Fatigue severity scale	49.52 ± 10.99	49.39 ± 11.69	47.47 ± 10.08	47.26 ± 11.33	−0.13	1.000	−3.92 to 3.66	−1.91	1.000	−5.70 to 1.88	−0.22	1.000	−4.01 to 3.57

* *p* < 0.05 using ANOVA of repeated measures (post hoc Bonferroni Adjustment). SD, standard deviation. Baseline measurement is before the first intervention period A, the second measurement is after the first intervention period A and before the intervention period B, the third measurement is after the intervention period B and before the second intervention period A, and the fourth measurement is after the second intervention period A.

**Table 3 jcm-11-00567-t003:** The statistical analysis of spatio-temporal parameters.

	Mean (SD)				Second vs. Baseline Measurement			Third vs. Second Measurement			Fourth vs. Third Measurement		
	Baseline	Second Measure	Third Measure	Fourth Measure	DM	*p*	CI 95%	DM	*p*	CI 95%	DM	*p*	CI 95%
Stride length	0.98 ± 0.23	0.98 ± 0.24	1.09 ± 0.23	1.08 ± 0.21	−0.0012	1.000	−0.07 to 0.06	0.113	<0.01 *	0.05 to 0.18	−0.01	1.00	−0.08 to 0.05
Single support	0.47 ± 0.08	0.47 ± 0.08	0.46 ± 0.10	0.48 ± 0.11	−0.0002	1.000	−0.03 to 0.03	−0.004	1.00	−0.03 to 0.02	0.02	0.70	−0.01 to 0.04
Double support	0.43 ± 0.26	0.43 ± 0.26	0.37 ± 0.21	0.41 ± 0.32	−0.0070	1.000	−0.07 to 0.05	−0.059	0.047 *	−0.12 to −0.0004	0.04	0.30	−0.02 to 0.10
Velocity	0.77 ± 0.24	0.76 ± 0.26	0.88 ± 0.29	0.86 ± 0.28	−0.0049	1.000	−0.08 to 0.07	0.114	0.001 *	0.04 to 0.19	−0.02	1.00	−0.09 to 0.06

* *p* < 0.05 using ANOVA of repeated measures (post hoc Bonferroni Adjustment). SD, standard deviation. Baseline measurement is before the first intervention period A, the second measurement is after the first intervention period A and before the intervention period B, the third measurement is after the intervention period B and before the second intervention period A, and the fourth measurement is after the second intervention period A.

**Table 4 jcm-11-00567-t004:** The statistical analysis of kinematic parameters.

Kinematic (Degrees)	Mean (SD)				Second vs. Baseline Measurement			Third vs. Second Measurement			Fourth vs. Third Measurement		
	Baseline	Second Measure	Third Measure	Fourth Measure	DM	*p*	CI 95%	DM	*p*	CI 95%	DM	*p*	CI 95%
Hip ROM	40.18 ± 8.21	40.25 ± 8.28	42.92 ± 6.69	43.20 ± 7.28	0.07	1.00	−2.03 to 2.18	2.66	0.006 *	0.56 to 4.77	0.28	1.00	−1.82 to 2.39
Hip angle at IC	30.75 ± 8.93	30.55 ± 8.89	32.07 ± 8.10	31.98 ± 8.38	−0.20	1.00	−2.54 to 2.14	1.52	0.47	−0.83 to 3.86	−0.24	1.00	−2.601 to 2.13
Peak hip extension stance period	−7.13 ± 9.69	−7.48 ± 10.04	−8.44 ± 8.97	−8.75 ± 7.56	−0.35	1.00	−3.16 to 2.46	−0.95	1.00	−3.76 to 1.86	0.60	1.00	−2.35 to 3.56
Peak hip flexion swing period	32.70 ± 9.12	32.42 ± 9.16	34.08 ± 8.45	34.18 ± 8.30	−0.28	1.00	−3.10 to 2.53	1.67	0.67	−1.15 to 4.48	−0.06	1.00	−2.91 to 2.78
Knee ROM	49.80 ± 4.18	49.74 ± 14.08	52.75 ± 12.18	51.79 ± 11.50	−0.06	1.00	−3.47 to 3.36	3.01	0.045 *	−0.40 to 6.43	−0.97	1.00	−4.38 to 2.45
Knee angle at IC	6.53 ± 8.32	6.35 ± 8.43	3.94 ± 7.69	4.78 ± 7.09	−0.19	1.00	−2.65 to 2.28	−2.23	0.097	−4.70 to 0.23	0.67	1.00	−1.80 to 3.14
Peak knee flexion swing period	50.04 ± 3.21	49.77 ± 13.08	49.66 ± 11.95	49.38 ± 11.32	−0.27	1.00	−4.20 to 3.65	−0.11	1.00	−4.03 to 3.82	−0.28	1.00	−4.20 to 3.65
Ankle ROM	26.94 ± 9.82	25.69 ± 7.11	26.00± 6.73	26.12 ± 5.19	−1.25	1.00	4.38 to 1.89	0.31	1.00	−2.83 to 3.44	0.12	1.00	−3.02 to 3.25
Ankle angle at IC	2.31 ± 6.64	1.85 ± 6.81	2.55 ± 5.63	2.65 ± 7.34	−0.46	1.00	−1.91 to 0.98	0.70	1.00	−0.74 to 2.15	0.10	1.00	−1.35 to 1.54
Peak ankle plantar flexion toe-off (º)	−3.14 ± 12.26	−2.05 ± 9.84	−3.03 ± 8.61	−3.30 ± 9.61	1.10	1.00	−1.41 to 3.60	−0.98	1.00	−3.49 to 1.52	−0.27	1.00	−2.77 to 2.23

ROM, range of motion; º, grade; IC, initial contact. * *p* < 0.05 using ANOVA of repeated measures (post hoc Bonferroni Adjustment). SD, standard deviation. Baseline measurement is before the first intervention period A, the second measurement is after the first intervention period A and before the intervention period B, the third measurement is after the intervention period B and before the second intervention period A, and the fourth measurement is after the second intervention period A.

## Data Availability

The data is available in the Movement Analysis, Biomechanics, Ergonomics and Motor Control Laboratory database. Data protection and the local Ethics Committee do not allow the data to be openly available. All analyzed data are presented in the article.

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
