# Peer review of "Reflex Locomotion Therapy for Balance, Gait, and Fatigue Rehabilitation in Subjects with Multiple Sclerosis"

_jcm, 2022, doi:10.3390/jcm11030567_

Round 1
Reviewer 1 Report
Your very good work is appreciated, and the findings and results are helpful and welcome. This reviewer however, finds a few aspects in your paper that require discussion. Would encourage comment in the manuscript about the limitations and facts posed below:
- The study was not controlled.
- The number of subjects was rather small.
- Agree entirely that patients with severe spasticity and ataxia, or combination, should be studied in further studies. These three aspects are in consonance with your statement that further research is granted.
- From the point of view of patient selection, it is unusual you utilized the McDonald Criteria 2001. The criteria has been updated several times: 2005, 2010 and the most recent version currently internationally used is the 2017.
- There is also need for an explanation for the utilization of the MMSE. Probably for the reasons of adequate mentation to undestand the study would be reasonable, but clarification that the MMSE is not acceptable in MS to assess cognition, is required. There is plenty of literature in this respect.
- Correct "RFL" in line 25 of the abstract. It should be RLT. Complete line 82 in the mauscript draft addressing the Research Ethics Committee of the "XX University".
Author Response
Revisor 1
- The study was not controlled. Thank you for your comment, the experimental protocol was designed based on a reversal design (Single-subject research), also called the ABA, with three intervention periods. We have reported this information in the methods section.
- The number of subjects was rather small. We obtained the effect size using the Software G*power (version 3.1.7) according to the statistical parameters described in the methods section. The parameters used are adequate for calculating the sample size according to the literature. However, we have reported in the study´s limitation this issue.
- Agree entirely that patients with severe spasticity and ataxia, or combination, should be studied in further studies. These three aspects are in consonance with your statement that further research is granted. Thank you for your comment
- From the point of view of patient selection, it is unusual you utilized the McDonald Criteria 2001. The criteria has been updated several times: 2005, 2010 and the most recent version currently internationally used is the 2017. We used the McDonald Criteria 2010, because de study was designed before the last version published in 2017. We have modified it in the methods section.
- There is also need for an explanation for the utilization of the MMSE. Probably for the reasons of adequate mentation to understand the study would be reasonable, but clarification that the MMSE is not acceptable in MS to assess cognition, is required. There is plenty of literature in this respect. Thank you for your comment, the appreciation is very useful. We have used MMSE to make sure that the patients understood the verbal instructions during the treatment and evaluation.
- Correct "RFL" in line 25 of the abstract. It should be RLT. Complete line 82 in the manuscript draft addressing the Research Ethics Committee of the "XX University". Thank you, we have changed it.
Reviewer 2 Report
In this manuscript authors analyzed Spatio-temporal parameters and kinematic variables of hip, knee, and ankle joints. Additionally, the Client Satisfaction Questionnaire (CSQ-8) was administered.
The main topic is of interest, however, I have some concerns
Revise extensively English with a mother tongue
Introduction is too poor
Add more information on fatigue and invisible symptoms in ms (fatigue, dysphagia, urinary/sexual dysfunction, bowel disfunction, cognitive impairment (consider DOI: 10.1097/01376517-200804000-00007; DOI10.1080/14656566.2020.1767068; doi: 10.1007/s00415-021-10737-w; doi.org/10.1016/j.jns.2005.07.018 )
Author Response
- Revise extensively English with a mother tongue. Thank you, we have reviewed.
- Introduction is too poor. Add more information on fatigue and invisible symptoms in ms (fatigue, dysphagia, urinary/sexual dysfunction, bowel disfunction, cognitive impairment (Consider DOI: 10.1097/01376517-200804000-00007; DOI10.1080/14656566.2020.1767068; DOI: 10.1007/s00415-021-10737-w; DOI: 10.1016/j.jns.2005.07.018). Thank you for your recommendations, we have included some information about these in the manuscript.
Reviewer 3 Report
abstract:
- “This study evaluateS”
- What does RFL stand for?
- What does ROM stand for?
Figure 1:
- Subfigure (d) feels like a paparazzi photo. Can’t you just take a normal picture and blur the face?
Methods
- A visual plot of the actual paradigm would be helpful
Results:
- What is the physical disability in the MS cohort? Why are no EDSS scores provided?
- The “difference of means” is meaningless; an effect size like Cohen’s d would be more informative
- I find the p-values rather surprising, certainly given that a Bonferroni correction has been applied. Looking e.g. at BBS, Baseline values are 46.21 +- 7.2 and third measure is 49.6+-5.6, I am very doubtful that this could lead to a p-value <0.001 given the relatively small effect size and the relatively small group. Could the authors share the code? Or plot the data in a figure?
- Is the observed improvement similar for all subjects?
- The effect seems sustained throughout the experiment (none of the parameters seem to go down again after the intervention). How do the authors explain this? How can we be sure that the effect is not mere a delayed effect of the first training period? Or is present only when combining the first two training periods?
- Is there no way to add a control group? E.g. why didn’t the authors include a B-A-B design? Or an A-A-A-A design as control?
Discussion:
- The authors may choose to omit the instruction paragraph at the end
- The discussion is rather limited and poorly connect to the rest of the literature.
Author Response
Abstract:
- “This study evaluateS” Thank you, we have changed it.
- What does RFL stand for? It´s an error. We have changed it.
- What does ROM stand for? We have added range of motion.
Figure 1:
Subfigure (d) feels like a paparazzi photo. Can’t you just take a normal picture and blur the face?
Your comment it´s funny but is the normal reaction when we apply the second phase of turning: elbow flexion, wrist dorsiflexion and extension and finger opening.
Methods
- A visual plot of the actual paradigm would be helpful. We have added the figure 1.
Results:
- What is the physical disability in the MS cohort? Why are no EDSS scores provided? We have added this information.
- The “difference of means” is meaningless; an effect size like Cohen’s d would be more informative. We have added the Cohen´s d of the significant parameters in the results. We have added information about it in the discussion section.
- I find the p-values rather surprising, certainly given that a Bonferroni correction has been applied. Looking e.g. at BBS, Baseline values are 46.21 +- 7.2 and third measure is 49.6+-5.6, I am very doubtful that this could lead to a p-value <0.001 given the relatively small effect size and the relatively small group. Could the authors share the code? Or plot the data in a figure? We have verified the result by performing a student's t test for related samples to compare the parameters between the third and the second measurements. According to the analysis, a t = 4.73 and a p-value = 0.000101 are obtained. This result is consistent with the data obtained through the Bonferroni a posteriori adjustment.
- Is the observed improvement similar for all subjects? There is some variability between subjects, mainly in the first measurements. After the RLT intervention, the observed variability decreases, as shown by some of the standard deviation values of the variables. In any case, the patients changed similarly.
- The effect seems sustained throughout the experiment (none of the parameters seem to go down again after the intervention). How do the authors explain this? How can we be sure that the effect is not mere a delayed effect of the first training period? Or is present only when combining the first two training periods? Indeed, the study design make difficult to explain the results. We have indicated this in the limitations. However, the intensity of therapy (session duration and frequency) was the same in all the intervention periods. Therefore, we could assume that the inclusion of RLT in the intervention period B motivated a clinical change in the patients that could have remained over time. We understand that the design is not the most appropriate, but it is difficult to carry out certain designs with better methodological quality in clinical settings.
- Is there no way to add a control group? E.g. why didn’t the authors include a B-A-B design? Or an A-A-A-A design as control? This is a good comment, but we have tried to ensure that all participants received a similar protocol for ethical reasons. We have added in the limitations of the study that future works should consider other designs.
Discussion:
- The authors may choose to omit the instruction paragraph at the end. Thank you, we apologize for the mistake.
- The discussion is rather limited and poorly connect to the rest of the literature. We have tried to improve this section.
Reviewer 4 Report
Clear and well written work, despite the small sample size. I suggest a revision of the discussion, some data are to be listed in the results too. Less redundancy might be appreciate. side effect reported are mild and I suppose patients do not needed any pharmacological support or other approach, please add some comments. Respiratory rate, if disposable, could be correlated to HR
Author Response
Clear and well written work, despite the small sample size. Thank you very much for your positive comments.
I suggest a revision of the discussion, some data are to be listed in the results too. Less redundancy might be appreciated. We have modified the result and discussion sections.
Side effect reported are mild and I suppose patients do not needed any pharmacological support or other approach, please add some comments. Respiratory rate, if disposable, could be correlated to HR. There were no major side effects. In fact, the participants showed a high satisfaction with the intervention. Regarding to the pharmacological treatment, all of patients continued with their medication. We did not measure the heart rate or the respiratory rate, therefore, we cannot comment about this issue.
Round 2
Reviewer 1 Report
Thank you for clarifying aspects requested by the reviewer.
Author Response
Thank you very much for your comments
Reviewer 3 Report
I would kindly like to ask to plot the data in Table 2. This is a relatively straightforward task and will increase the trust in the present statistics.
Author Response
Thank you very much for your comments. We believe that it has helped us to improve the manuscript. We have included figure 3 with table 2 data